# Determinants of early initiation of first antenatal care visit in Ethiopia based on the 2019 Ethiopia mini-demographic and health survey: A multilevel analysis

Gossa Fetene Abebe[1]*, Melsew Setegn Alie[2], Desalegn Girma[1], Gosa Mankelkl[2], Ashenafi Assefa Berchedi[2], Yilkal Negesse[3]

1 Department of Midwifery, College of Medicine and Health Sciences, Mizan-Tepi University, Mizan, Ethiopia, 2 College of Medicine and Health Sciences, Mizan-Tepi University, Mizan, Ethiopia, 3 College of Health Science, Debre-Markos University, Debre-Markos, Ethiopia

* Feteneg2119@gmail.com

## Abstract

### Background

Early initiation of the first antenatal care visit provides a critical opportunity for health promotion, disease prevention, and curative care for women and their unborn fetuses. However, in developing countries, including Ethiopia, it is underutilized and most of the pregnant women didn't attend antenatal care visits during the first trimester (early). Therefore, the objective of this study was to estimate the prevalence of early initiation of antenatal care visits and its determinants among reproductive-age women in Ethiopia.

### Methods

A secondary data analysis was done based on the 2019 intermediate Ethiopian demographic health survey. The data were weighted by sampling weight for probability sampling and non-response to restore the representativeness of the data and have valid statistical estimates. Then, a total weighted sample of 2,935 women aged 15–49 years who gave birth in the five years preceding the survey and who had antenatal care visits for their last child was included. A multilevel mixed-effects logistic regression model was fitted to examine the determinants of early initiation of first antenatal care visits. Finally, statistical significance was declared at a p-value < 0.05.

### Results

In this study, the overall magnitude of early initiation of the first antenatal care visit was 37.4% (95%CI: 34.6–40.2%). Women who attend higher education (AOR = 2.26: 95%CI; 1.36–3.77), medium wealth status (AOR = 1.80: 95%CI; 1.17–2.76), richer wealth status (AOR = 1.86: 95%CI; 1.21, 2.85), richest wealth status (AOR = 2.34: 95%CI; 1.43–3.83), living in Harari region (AOR = 2.24: 95%CI; 1.16–4.30), and living at Dire-Dawa city (AOR = 2.24: 95%CI; 1.16–4.30) were higher odds of early initiation of first ANC visits. However,

Data Availability Statement: The third-party data was obtained for this study from the DHS program. Data are available from the measure DHS website

http://www.dhsprogram.com for researchers who meet the criteria for access to confidential data. Furthermore, the researchers can access the birth record file of the EDHS dataset from https://www.dhsprogram.com/data/dataset_admin/login_main.cfm after securing written consent from the DHS website.

**Funding:** The authors received no specific funding for this work.

**Competing interests:** The authors have declared that no competing interests exist.

**Abbreviations:** ANC, Antenatal Care; AOR, Adjusted odds ratio; CI, Confidence Interval; HIV, Human Immunodeficiency Virus; SNNPR, South Nation Nationality Peoples Region; WHO, World Health Organization.

women who were rural resident (AOR = 0.70: 95%CI; 0.59–0.93), household headed by male (AOR = 0.87: 95%CI; 0.72, 0.97), having $\geq$ 5 family size (AOR = 0.71: 95%CI; 0.55–0.93), and living in SNNPRs (AOR = 0.44: 95%CI; 0.23–0.84) were lower odds of early initiation of first ANC visits.

## Conclusion

The prevalence of early initiation of first antenatal care remains low in Ethiopia. Women's education, residence, wealth status, household head, having $\geq$ 5 family sizes, and region were determinants of early initiation of first antenatal care visits. Improving female education and women's empowerment through economic transitions with special attention given to rural and SNNPR regional state residents could maximize the early initiation of first antenatal care visits. Furthermore, to increase early antenatal care uptake, these determinants should be considered when designing new policies or updating policies and strategies on antenatal care uptake to help increase early attendance, which can help in the reduction of maternal and neonatal mortality and to achieve sustainable development goals 3 by 2030.

## Introduction

Maternal and neonatal health has gained a global health priority in the past decades, exemplified by its adoption as the fourth and fifth Millennium Development Goals (MDG) [1, 2] and continued in the third Sustainable Development Goals (SDG) [3]. Nevertheless, the goal to reduce maternal mortality by 75% in the year 2015 was not successful and lagged behind its target among all [1]. Regardless of the 2015 commitment, about 295,000 women died during and following pregnancy and childbirth in 2017 [4] globally. Sub-Saharan Africa and Southern Asia shoulder the highest-burden (86%) of this estimated global maternal deaths, at which Sub-Saharan Africa alone accounted for two-thirds (196, 000) of maternal deaths [4], where the health systems are weak in access and minimum health services utilization [5]. The maternal mortality ratio (MMR) in Ethiopia, one of the Sub-Saharan African countries, is 412 maternal deaths per 100,000 live births. Also, globally around 2 million stillbirths [6] in the year 2019 and 2.4 million neonatal deaths [7] in the year 2020 were reported. Of which the highest numbers were contributed by Sub-Sahan Africa including Ethiopia and Southern Asia [6, 7].

Sustaining availability, accessibility, and providing quality maternal healthcare services for all pregnant women at all levels is crucial to optimizing the maternal and newborn healths status [8–10]. Antenatal care is one of the proven maternal care services strategies for reducing maternal and neonatal morbidity and mortality directly by the identification and treatment of pregnancy-related illness, or indirectly by detection of women at risk of complications of delivery and counseled them to deliver in an appropriately equipped health facility [11]. Timely (initiated at the first trimester of pregnancy) [12, 13], frequent (four or more visits) [12, 14, 15], and adequate (with proper contents) [12, 14, 15] antenatal care services provided for the mother during the period of pregnancy reduce the risk of complication and death for the mother as well as the unborn fetus [9] and improve the uptake of subsequent maternal health services. The time at which the first antenatal care services were initiated has the utmost merits to ensure optimal health impacts for both woman and their unborn fetus [11]. According to the 2016 World Health Organization (WHO) antenatal care recommendation for positive pregnancy outcomes, the first antenatal care visits should be within the first trimester (early)

[11]. However, the magnitude of early antenatal care visits is very low (24%) in developing countries as compared to developed countries (81.9%) [16]. Shreds of evidence noted that many pregnant women attend their first ANC visits late (after the first trimester) [17–19]. A systematic review study done in Ethiopia by Tesfaye G. et al. identified that the majority (64%) of pregnant women start their first ANC visits late [20].

Early initiation of ANC visits is one of the pillar components of ANC services that aimed to have baseline information on the general well being of the pregnant women, accurate gestational age ascertainment, screening of pre-existing problems of the women such as human immunodeficiency virus (HIV), syphilis, hepatitis, malaria, anemia, and other chronic medical disorders, and early detection of complications arising during pregnancy [11, 16]. Timely screening and providing appropriate therapy of HIV and syphilis help to halt mother to fetus transmission. If the mothers are left untreated early, a 70–100% probability of transmitting the infection to their unborn fetus, and one-third of pregnancies will end up with stillbirth [21, 22]. Furthermore, early initiation of ANC is a good entry point to discuss birth preparedness and complication readiness plan and to augment awareness of its sign and symptoms between pregnant women and health care providers [17]. It also creates the opportunity to provide immunizations against tetanus, supplementation of iron and folic acid to prevent anemia and neural tube defect, counseling on nutrition, and malaria and worms prophylactic treatments [11]. Early initiation of antenatal care services also has a positive impact on the decrement of poor perinatal outcomes such as low birth weight, preterm birth, and jaundice [23, 24]. In summary, early ANC visits aim to screen complications or predictors for the occurrence of complications which enable timely interventions to handle the negative impacts of such complications on a pregnant woman and unborn fetus [25].

To improve the uptake of ANC services early, the government of Ethiopia in collaboration with other non-governmental organizations (NGOs) has implemented different initiatives that expand access such as primary health care expansion, health extension programs, and charge-free maternal health services [26–30]. Despite, all these efforts made, in Ethiopia, the early initiation of first ANC visits during the first trimester as recommended by WHO [11] is still low, and most pregnant women attend their first ANC visits late [20]. As reported in the Ethiopian Demographic and Health Survey (EDHS) 2016, only 20% of pregnant women were attend their first ANC visit in the first trimester of pregnancy [31] as recommended by WHO [11]. This indicated that there are determinants that became bottlenecks for the increment of early initiation of first ANC visits and need further investigation.

Previously studies done in Ethiopia identified that early initiation of first ANC visits was impending by different determinants such as place of residence [20, 32, 33], maternal educational status [20, 32, 33], husband education status [34], maternal occupation [20], maternal age [20, 33], marital status, household wealth-income [20, 33], parity [20, 32], partner involvement [20], pregnancy intention [20, 32, 33], knowledge on antenatal care service [20, 34], means of approving current pregnancy [20, 34], being advised before starting antenatal care visit [34], exposure to mass media [33], pregnancy complication [20], having a history of abortion or stillbirth [35], covered by health insurance, distance from health facilities [33], and regions [32].

However, most of the previous studies in Ethiopia were conducted in specific areas with small sample sizes, mainly facility-based, and were not nationally representative. Moreover, the previous studies mainly emphasized on individual-level determinants with little attention given to community-level determinants. However, this could underscore the importance of considering contextual determinants when designing appropriate antenatal care service strategies. Thus, for bridging all those gaps, this study used a recent national-level data (2019 Ethiopia Mini Demographic and Health Survey data) to determine the current nationwide

magnitude, and individual and community-level determinants of early initiation of the first antenatal care visit among reproductive-age women. Therefore, the findings of this study will be invaluable to identify the determinants that impede the early initiation of first ANC visits as a result respective programs are shaped more appropriately towards the identified determinants and will also have paramount importance to show country-level figures, and screen out modified and persistent determinants, which in turn maximize the timely initiation of the first ANC visits, and achievement of the SDG #3 of eliminating maternal and neonatal mortality by 2030.

## Methods and materials

### Study setting and period

Secondary data analysis was employed based on the 2019 Ethiopian mini-demographic and health survey data, which were done from 21st March/2019 to 28th June/2019. To collect the data, a cross-sectional study design was employed. Ethiopia is located in the horn of Africa, between 3˚ - 15˚ North latitude and 33˚ - 48˚ East longitudes. It has nine regions and two city administrations. Ethiopia has been conducting Demographic and health surveys (EDHS) started in the year 2000 and then conducted every five years. Two Ethiopian Mini Demographic and Health Surveys (EMDHS) in 2014 and 2019 have been employed. The EMDHS is usually conducted in between the standard EDHS.

### Data source/extraction

After permission was secured through an online request by explaining the aim of the study, the data were taken from the Measure Demographic and Health Surveys (DHS) website (http://www.dhsprogram.com/).

### Population of the study

The source population of the study was all reproductive age women (15–49 years) who gave birth in the five years preceding the survey and who had at least one ANC visit for their last child all over Ethiopia, whereas women who gave birth in the five years preceding the survey and who had at least one ANC visit for their last child and lived in the selected enumeration areas were the study populations.

### Eligibility criteria

In this study, all reproductive age women who gave birth in the five years preceding the survey and found in the selected clusters at least one night before the data collection period were included, whereas, women who had no ANC visit and unknown first date of ANC visit were excluded. Accordingly, a total of 2,935 weighted samples of reproductive age women were incorporated (**Fig 1**).

### Sampling technique

For all EDHS, a two-stage stratified cluster sampling technique was used. In the first stage, stratification was done by region, and then each region was stratified as urban and rural. In the 2019 EMDHS data, a total of 305 (94 urban, and 211 rural) enumeration areas (EAs) were selected using probability proportional to EA size in in the first stage. In the second stage, households were selected proportionally from each EA by using a systematic sampling method. The detailed method of data collection was accessed at the DHS database [36].

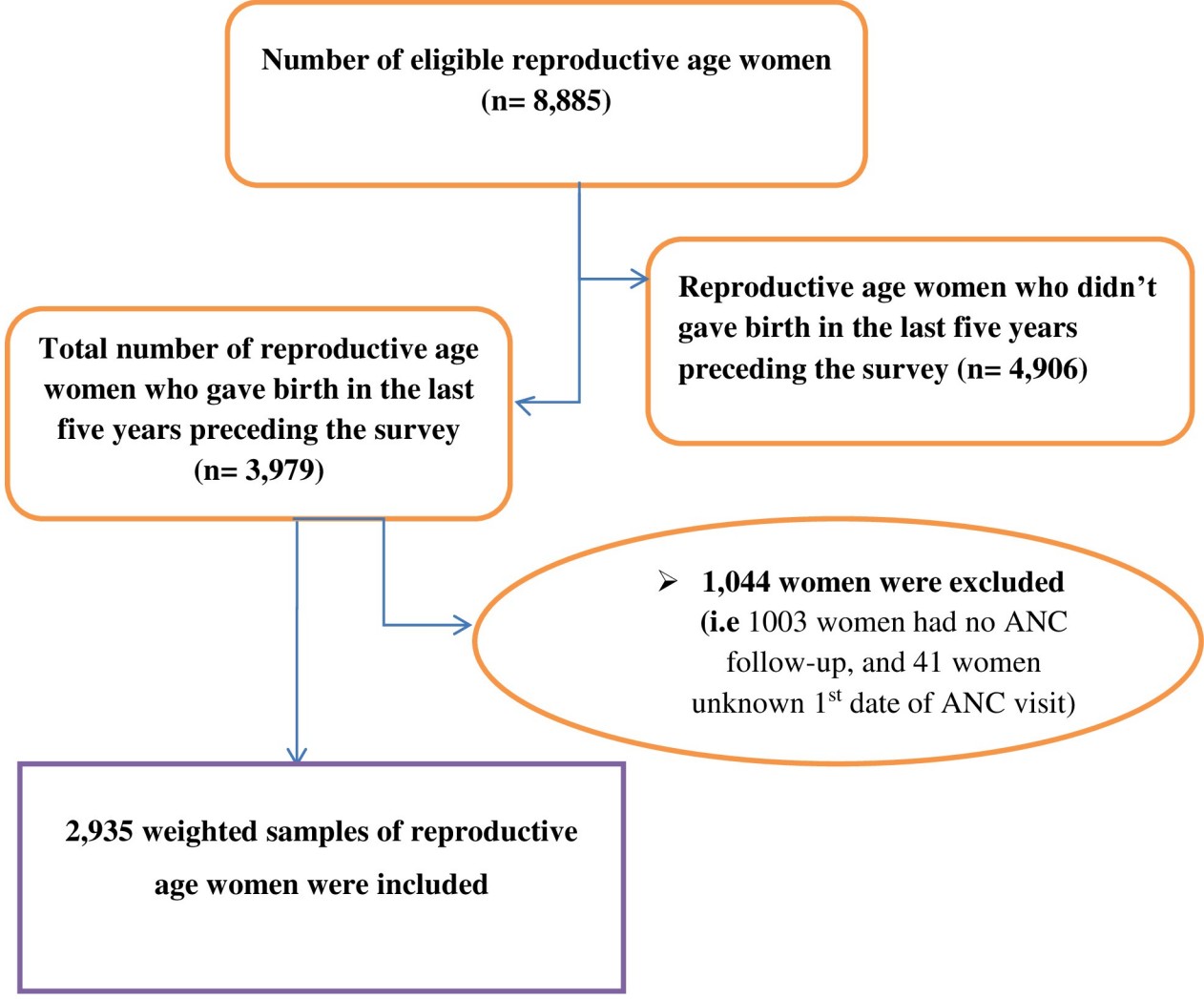

**Fig 1. Schematic presentation showing the sampling and exclusion procedures to identify the final sample size in 2019 EMDHS.**

### Variables of the study

**Dependent variable.** The dependent variable was early initiation of first ANC visit which was a binary outcome variable and classified as "early" if a woman attended ANC visit within the first 12 weeks of gestation (coded as "Yes = 1"), and "late" if she attended ANC visit after 12 weeks of gestation (coded as "No = 0") [11].

**Independent variables.** The independent variables were classified as individual and community-level variables. The individual-level variables include maternal age, marital status, religion, educational status of women, household wealth status, parity, preceding birth interval, the contraceptive method used, sex of household head, and the number of family size. Whereas, the place of residence and region of the study participants were considered as community-level variables. The detailed information's about the explanatory factors were presented in **Table 1**.

### Statistical analysis

The data were extracted from the individual record (IR) file data set using SPSS version 24 software and further analyzed using STATA version 15. By using sample weight the data were

**Table 1. The description of some of the independent variables.**

| Variables | Definitions/Categories |
|---|---|
| Age of the mothers (in years) | The age of the mother was coded as 15–19, 20–24, 25–29, 30–34, 35–39, 40–44, and 45–49; however, during data cleaning some of the categories lack or have very limited participants, thus, we regrouped into 15–19, 20–29, 30–39, and 40–49 |
| Religion | Religion was coded as Protestant, Orthodox, Muslim, Catholic, traditional, other. Depending on the number of participants, we sum up Catholic, and Traditional, as "Others". And, we regrouped into Protestant, Orthodox, Muslim, and others |
| Marital status | Marital status was coded as married, never in the union, living with partner, widowed, divorced, and no longer living together/separated; but, participants from the last five were very few, so, we categorized married and unmarried by living married alone and classified all else. |
| Birth order | Birth order was a count number ranging from 1 to 15; however, we regrouped into 1$^{st}$, 2$^{nd}$, and third and above |
| Parity | The total number of children ever born was a count number ranging from 1 to 15; however, we regrouped into 1$^{st}$, and 2$^{nd}$ and above |
| Preceding birth interval | Preceding birth interval was coded from 1 to 219 months in the dataset; however, during data cleaning some of the categories lack or have very limited participants, thus, we regrouped into < 24, 24–36, and >36 months. |
| Number of a household member | The number of household size was a count number ranging from 1 to 24; however, during data cleaning some of the categories lacked or have very limited participants, so, we recoded into $\leq 4$ and $\geq 5$. |
| Contraception | EMDHS grouped the variable No-method, Traditional method, Modern method |

weighted for probability sampling and non-response to restore the representativeness of the survey and get reliable statistical estimates. Data editing, cleaning, and coding were done. Descriptive statistics were done and presented by tables. Socio-demographic and other profiles of the study participants were compared using the chi-square test. In this study, two levels of data hierarchy were considered due to the sampling technique applied in EDHS (Multistage stratified cluster sampling). Level one unit was the individual pregnant woman in the households and level two units were enumeration areas. Level one (pregnant women in the households) was nested within units at the next higher level (enumeration areas). The outcome variable was represented by $Y_{ij} = \begin{cases} \textit{Early initiation of ANC} \\ \textit{Late initiation of ANC} \end{cases}$, the category is dichotomous.

Therefore, the multilevel mixed effects logistic regression model was fitted to identify the factors influencing early initiation of first antenatal care services at each level (individual level and community level). Four models were fitted for the multilevel logistic regression. The first model (a model without covariate) was fitted to determine the extent of cluster variation in the early initiation of ANC visits. The second model was fitted with individual-level factors alone. The third model was fitted with community-level variables. Lastly, the fourth model was fitted with both individual and community-level factors. Both bivariable and multivariable analyses were employed. Variables that have a P-value of $\leq 0.25$ in the bivariable multi-level logistic regression analysis were candidates for multivariable multilevel logistic regression analysis. Then Variables in multilevel multivariable logistic regression were declared to be statistically significant at a P-value of < 0.05. The fitted models were compared based on Akaike's Information Criteria (AIC) and a model with a small AIC value was selected and all interpretations and inferences were made based on this model. The random-effects measure the variation of early initiation of ANC across clusters (EAs) and are determined using the Intraclass correlation coefficient (ICC), median odds ratio (MOR), and proportional change in variance (PCV) statistics. The ICC determines the variation within-cluster and between-cluster differences. The PCV determines the total variation of early initiation of ANC at the individual- and

community-level factors in each model. The MOR measures the MOR of early initiation of ANC at the high-risk cluster (clusters did not attend ANC early) and low-risk cluster (clusters having a high prevalence of early initiation of ANC) when we select randomly two pregnant women during data collection from two clusters. The formulas used to calculate these three measurements are as follows;

ICC = $v_i/(v_i + \pi^2/3) \sim \frac{V_i}{V_i+3.29}$, where $V_i$ = between cluster (community) variances and $\pi 2$ /3 = within-cluster (community) variance [37].

PCV = $\frac{V_i-V_y}{V_i}$, where Vi = variances of the null model, where Vy = variance of the model with more terms [37].

MOR = $exp.[\sqrt{2 \times Vz} \times 0.6745] \sim exp.[0.95\sqrt{Vz}]$ where Vz = variance at the community level [37].

### Ethical consideration

Ethical approval was obtained from measure Demographic Health Survey (DHS) after filling the requesting form for accessing the data. The data used in this study are freely available, aggregated secondary data that didn't contain any personal identifiers that can be linked to the study participants (http://www.dhsprogram.com). The requested data were used in strictly anonymous and served only for the study purpose. The full information about the ethical issue was available in the EMDHS-2019 report.

### Results

### Sociodemographic characteristics of the study participants

In this study, a total of 2,935 weighted samples of the study participants have participated from all over the country. The overall prevalence of early initiation of antenatal care was 37.4% (95% CI: 34.6–40.2). More than half (55.2%) of the study participants were in the age range of 20–29 (55.2%) years old. Of the total study participants, 2,034 (69.3%) were urban dwellers, 2,691 (91.7%) were married, 1,212 (41.29%) were Muslim religion followers, 1,274 (43.41%) didn't attend formal education, and 2,359 (79.69%) were from male-headed households (**Table 2**). The majority of the participants who didn't initiate first ANC visits early were from the SNNPR regional state (14.88%) followed by the Oromia regional state (11.93%) (**Fig 2**).

### Obstetric and reproductive related characteristics of reproductive-age women in Ethiopia, 2019

Of the total study participants, more than two-thirds (64.46%) of the households have had $\geq 5$ family members and nearly three fourth (75.9%) were multiparous. The majority (62.49%) of the women had given birth more than two times, 61.32% of the woman have had a birth spacing of 36 months and above, and 57.68% of the women didn't use contraception (**Table 3**).

### Random effect and model comparison

The Intraclass correlation coefficient (ICC) in the null model was (0.213), which means that 21.3% of the variability of early initiation of ANC was due to the differences between clusters or unobserved factors at the community level. This indicates that the multilevel logistic regression model is best to estimate early initiation of first antenatal care visits among pregnant women than single-level logistic regression. The Akaike's Information Criteria (AIC) is smallest at model 4 (AIC = 2,008.2) as compared to random intercept only model or null model (AIC = 3821.3), a model with only individual-level factors (AIC = 2038.8), and model with

**Table 2. Early initiation of first antenatal care visits by sociodemographic characteristics of reproductive-age women in Ethiopia, 2019.**

| Variable | Category | Weighted frequency (%) | Early initiation of ANC | |
|---|---|---|---|---|
| | | | No (%) | Yes (%) |
| Women age (years) | 15–19 | 170 (5.99) | 102 (6.32) | 68 (5.56) |
| | 20–29 | 1,567 (55.2) | 865 (53.59) | 702 (57.35) |
| | 30–39 | 999 (35.2) | 579 (35.87) | 420 (34.31) |
| | 40–49 | 102 (3.59) | 68 (4.21) | 34 (2.78) |
| Residence | Urban | 901 (30.7) | 360 (21.43) | 541 (43.11) |
| | Rural | 2,034 (69.3) | 1,320 (78.57) | 714 (56.89) |
| Marital status | Married | 2,691 (91.7) | 1,552 (92.38) | 1,139 (90.76) |
| | Unmarried | 244 (8.3) | 128 (7.62) | 116 (9.24) |
| Religion | Orthodox | 1,095 (37.31) | 584 (34.76) | 511 (40.72) |
| | Muslim | 1,212 (41.29) | 681 (40.54) | 531 (42.31) |
| | Protestant | 586 (19.97) | 388 (23.10) | 198 (15.78) |
| | Others[C] | 42 (1.43) | 27 (1.61) | 15 (1.20) |
| Maternal educational status | No education | 1,274 (43.41) | 843 (50.2) | 431 (34.3) |
| | Primary | 1,074 (36.59) | 602 (35.8) | 472 (37.2) |
| | Secondary | 359 (12.23) | 164 (9.8) | 195 (15.5) |
| | Higher | 228 (7.77) | 71 (4.2) | 157 (12.5) |
| Household wealth status | Poorest | 598 (20.27) | 423 (25.18) | 175 (13.94) |
| | Poorer | 503 (17.14) | 338 (20.12) | 165 (13.15) |
| | Middle | 458 (15.6) | 298 (17.74) | 160 (12.75) |
| | Richer | 461 (15.71) | 281 (16.73) | 180 (14.34) |
| | Richest | 915 (31.18) | 340 (20.24) | 575 (45.82) |
| Sex of household head | Female | 596 (20.31) | 307 (18.27) | 289 (23.03) |
| | Male | 2,339 (79.69) | 1,373 (81.73) | 966 (76.97) |

ANC: Antenatal care, [c] catholic or traditional religion follower

only community-level factors (AIC = 3702.2). Therefore this model is the best-fitted model for the data because it has the smallest AIC as compared to the rest models. Therefore, all interpretations and reports were made based on this model. In addition, the median odds ratio (MOR) in all models was greater than one noted that there is a variation in early initiation of ANC among pregnant women between community levels. The value of MOR (2.44) in the null model depicts that there was a variation of early initiation of ANC between clusters when we randomly select pregnant women from two clusters, women from clusters that have a high prevalence of early initiation of ANC were 2.44 times more likely to attend ANC early as compared to women from low prevalence of ANC utilization cluster. The higher proportional change in variance (PCV) value in the fourth model (0.73) showed that about 73% of the variability of early initiation of ANC visits was explained by both the individual-level and community-level factors (**Table 4**).

## Determinants of early initiation of first antenatal care among reproductive-age women in Ethiopia, 2019

In the final model (model 4) both individual and community-level factors were added for multilevel analysis, of which, maternal educational status, household wealth status, sex of household head, family size, residence, and region were significantly associated with early initiation of antenatal care visits in Ethiopia at P-value < 0.05 (Table 4). The odds of early initiation of

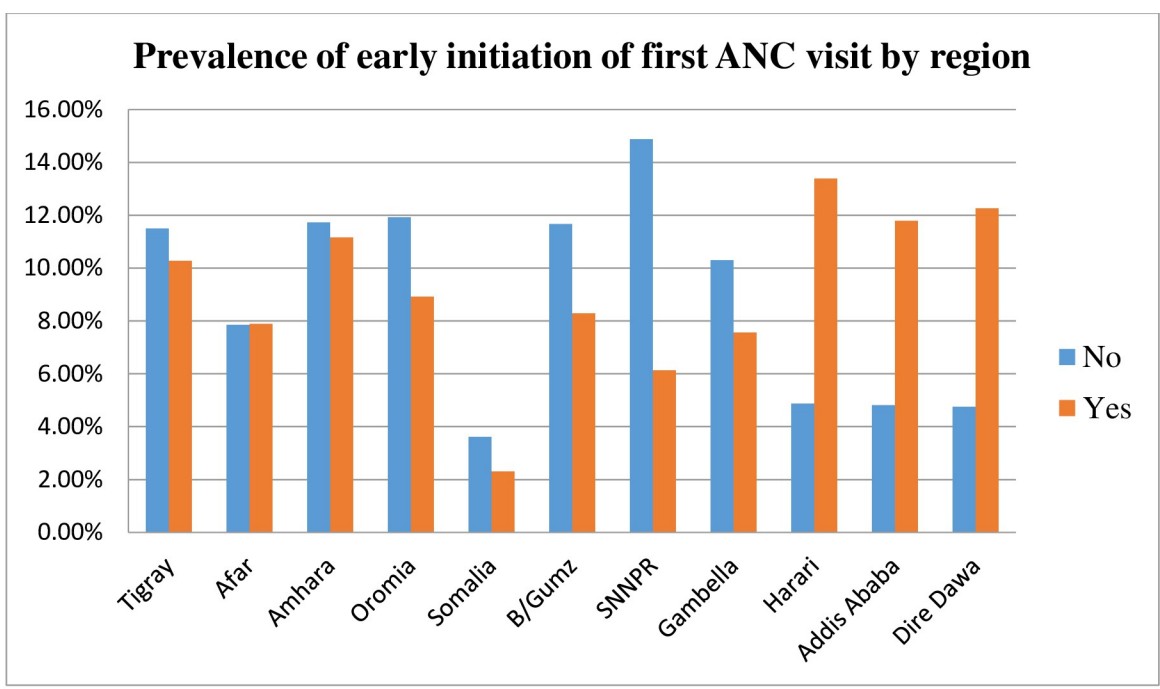

Key; B/Gumz- Benshangule Gumz, SNNPR- South Nation Nationalities Peoples Region

**Fig 2. The prevalence of early initiation of first ANC visits by regions in Ethiopia, 2019.**

**Table 3. Early initiation of antenatal care by different obstetric and reproductive health-related characteristics of reproductive-age women in Ethiopia, 2019.**

| Variable | Category | Weighted frequency (%) | Early initiation of ANC | |
|---|---|---|---|---|
| | | | No (%) | Yes (%) |
| Parity | Primiparous | 707 (24.1) | 344 (20.48) | 363 (28.92) |
| | Multiparous | 2,228 (75.9) | 1,336 (79.52) | 892 (71.08) |
| Birth order | First | 1,146 (19.6) | 225 (18.89) | 171 (19.79) |
| | Second | 1,047 (17.91) | 228 (19.14) | 140 (16.20) |
| | Thrid and above | 3,653 (62.49) | 738 (61.96) | 553 (64) |
| Preceding birth interval (months) | < 24 | 285 (6.09) | 49 (5.11) | 40 (5.77) |
| | 24–36 | 1,525 (32.59) | 304 (31.7) | 196 (28.28) |
| | >36 | 2,869 (61.32) | 606 (63.19) | 457 (65.95) |
| Contraceptive method used | No-method | 1,693 (57.68) | 1,011 (60.18) | 682 (54.34) |
| | Modern | 1,219 (41.53) | 657 (39.11) | 562 (44.78) |
| | Traditional | 23 (0.78) | 12 (0.71) | 11 (0.88) |
| Family member | ≤ 4 | 1,043 (35.54) | 536 (31.9) | 507 (40.4) |
| | ≥ 5 | 1,892 (64. 46) | 1,144 (68.1) | 748 (59.6) |
| Number of antenatal care | Once | 141 (4.80) | 119 (7.08) | 22 (1.75) |
| | Two times | 353 (12.03) | 282 (16.79) | 71 (5.66) |
| | Three times | 768 (26.17) | 526 (31.31) | 242 (19.28) |
| | Four and above | 1,673 (57) | 753 (44.82) | 920 (73.31) |

ANC; Antenatal care

Table 4. Random effect and model comparison for predictors of early initiation of antenatal care among reproductive-age women in Ethiopia, 2019.

| Parameter | Null (model I) | Model II | Model III | Model IV |
|---|---|---|---|---|
| ICC | 21.3% | 10.1% | 9.2% | 4.6% |
| Variance | 0.89 (0.66, 1.21) | 0.37 (0.19, 0.69) | 0.33 (0.21, 0.52) | 0.15 (0.05, 0.52) |
| MOR | 2.44 | 1.55 | 1.48 | 1.03 |
| PCV | Reference | 59% | 63% | 73% |
| Model fitness | | | | |
| AIC | 3821.3 | 2038.8 | 3702.2 | 2,008.2 |

AIC: Akaike's information criteria, ICC: Intraclass correlation coefficient, MOR: Median Odds ratio, PCV: proportional change in variance

antenatal care visits were two times (AOR = 2.26: 95%CI; 1.36–3.77) more likely among women who attended higher educations as compared to those women who didn't attend formal education. Women who belong to the household wealth status of medium, richer, and richest were 1.8 (AOR = 1.80: 95%CI; 1.17–2.76), 1.86 (AOR = 1.86: 95%CI; 1.21, 2.85), and 2.3 (AOR = 2.34: 95%CI; 1.43–3.83) times more likely to attend antenatal care visits as compared to women who belong to poorest wealth status, respectively. The odds of the utilization of early ANC was 13% (AOR = 0.87: 95%CI; 0.72, 0.97) less likely among women whose households were headed by a male as compared to their counterparts. Likewise, the odds of attending early ANC was 29% (AOR = 0.71: 95%CI; 0.55–0.93) less likely among women who had $\geq 5$ family size as compared to those women who have had less than five family size. Women who resided in rural area was 30% (AOR = 0.70: 95CI; 0.59–0.93) less likely to attend early antenatal care services than women who lived in urban. Women who lived in the Harari region and Dire-Dawa city administration were two (AOR = 2.24: 95%CI; 1.16–4.30), and three (AOR = 3.29: 95%CI; 1.69–6.38) times more likely to attend early antenatal care visits as compared with women lived in Tigray region. Whereas, women who lived in SNNPRs were 66% (AOR = 0.44: 95%CI; 0.23–0.84) less likely to attend early antenatal care visits as compared to their counterparts (**Table 5**).

## Discussion

Even though early initiation of antenatal care services is a pillar strategy for achieving good maternal and neonatal health outcomes through early detection and prevention of risks during pregnancy [38, 39], the magnitude of early initiation of ANC services is still alarmingly low in Ethiopia [40]. Therefore, this study provides up-to-date information about the magnitude and predictors of early initiation of antenatal care service in Ethiopia using the 2019 intermediate Ethiopia demographic health survey.

In this study, the magnitude of early initiation of ANC visits was 37.4% (95%CI: 34.6–40.2%). This result was in line with studies done in Ethiopia, for instance, a systematic review done by Gezahegn et. al., (36%) [20], Jimma University specialized hospital (39.9%) [41], and Gondar town (35.4%) [42]. However, this study was higher than studies conducted in Ethiopia like the study done by Teshale A. et. al., (32.69%) [32], Shebedino district (21.71%) [43], Debre Berhan health institutions (26.2%) [44], Kembata Tembaro Zone (31.4%) [35], and Debre Markos town (33.4%) [45]. This divergence might be due to the difference in the gaps in the study period [32, 35, 44], the launching and strengthened functioning of the Health Extension Program (HEP) and women development army, and improving access to health care systems in the country.

On the other hand, the magnitude of early initiations of antenatal care visits was lower than studies done in Ethiopia, for example, Addis Ababa (58%) [46], Bule Hora district (57.8%)

**Table 5. Multilevel logistic regression analysis to assess determinants of early initiation of first ANC visits among reproductive-age women in Ethiopia, 2019.**

| Variable | Category | Null model I | Model II<br>AOR (95%CI) | Model III<br>AOR (95%CI) | Model VI<br>AOR (95%CI) |
|---|---|---|---|---|---|
| Maternal age (Years) | 15–19 | - | 1 | - | 1 |
| | 20–29 | - | 0.98 (0.60, 1.61) | - | 1.01 (0.62, 1.65) |
| | 30–39 | - | 1.13 (0.65, 1.94) | - | 1.21 (0.71, 2.08) |
| | 40–49 | - | 1.15 (0.52, 2.53) | - | 1.30 (0.60, 2.83) |
| Marital status | Married | - | 1 | - | 1 |
| | Unmarried | - | 1.26 (0.67, 2.6) | - | 1.68 (0.98, 2.62) |
| Religion | Orthodox | - | 1 | - | 1 |
| | Muslim | - | 0.94 (0.70, 1.27) | - | 0.77 (0.53, 1.10) |
| | Protestant | - | 0.53 (0.37, 0.76) | - | 0.81 (0.53, 1.24) |
| | Others [c] | - | 1.44 (0.53, 3.89) | - | 2.05 (0.75, 5.57) |
| Educational status | No education | - | 1 | - | 1 |
| | Primary | - | 1.21(0.91, 1.59) | - | 1.32 (1.00, 1.74) |
| | Secondary | - | 1.30 (0.87, 1.95) | - | 1.35 (0.90, 2.02) |
| | Higher | - | 2.37 (1.41, 3.97) | - | 2.26 (1.36, 3.77)* |
| Household wealth status | Poorest | - | 1 | - | 1 |
| | Poorer | - | 1.60 (0.48, 1.38) | - | 1.52 (0.91, 1.69) |
| | Middle | - | 1.58 (1.04, 2.04) | - | 1.80 (1.17, 2.76)* |
| | Richer | - | 1.70 (1.11, 2.59) | - | 1.86 (1.21, 2.85)* |
| | Richest | - | 3.51 (2.34, 5.26) | - | 2.34 (1.43, 3.83)* |
| Total number of children ever born | One | - | 1 | - | 1 |
| | 2 and above | - | 0.99 (0.71, 1.36) | - | 0.99 (0.72, 1.37) |
| Birth interval (in months) | < 24 | - | 1 | - | 1 |
| | 24–36 | - | 0.73 (0.43, 1.24) | - | 0 .70 (0.42, 1.17) |
| | >36 | - | 0.84 (0.51, 1.39) | - | 0.82 (0.50, 1.36) |
| Contraceptive method used | No-method | - | 1 | - | 1 |
| | Modern | - | 1.17 (0.91, 1.49) | - | 1.24 (0.97, 1.59) |
| | Traditional | - | 1.84 (0.53, 6.38) | - | 2.02 (0.58, 7.06) |
| Sex of household head | Female | - | 1 | - | 1 |
| | Male | - | 0.66 (0.45, 0.79) | - | 0.87 (0.72, 0.97)* |
| Family size | ≤ 4 | - | 1 | - | 1 |
| | ≥ 5 | - | 0.72 (0.55, 0 .94) | - | 0.71 (0.55, 0.93)* |
| Community-level variables | | | | | |
| Residence | Urban | - | - | 1 | 1 |
| | Rural | - | - | 0.48 (0.36, 0 .63) | 0.70 (0.59, 0.93)* |
| Region | Tigray | - | - | 1 | 1 |
| | Afar | - | - | 1.05 (0.64, 1.72) | 1.58 (0.80, 3.12) |
| | Amhara | - | - | 1.08 (0.69, 1.70) | 1.21 (0.71, 2.05) |
| | Oromia | - | - | 0.72 (0.46, 1.14) | 0.59 (0.33, 1.07) |
| | Somalia | | - | 0.57 (0.30, 1.10) | 1.10 (0.47, 2.59) |
| | B/Gumz | | - | 0.80 (0.50, 1.30) | 0.71 (0.39, 1.28) |
| | SNNPR | | - | 0.45 (0.28, .72) | 0.44 (0.23, 0.84)* |
| | Gambella | | - | 0 .79 (0.48, 1.28) | 0.66 (0.35, 1.23) |
| | Harari | | - | 2.49 (1.51, 4.11) | 2.24 (1.16, 4.30)* |
| | Addis Ababa | | - | 1.73 (1.00, 2.98) | 1.32 (0.69, 2.52) |
| | Dire Dawa | | - | 2.33 (1.40, 3.88) | 3.29 (1.69, 6.38)* |

*$p$-value < 0.05

AOR: Adjusted odds ratio, CI: Confidence interval, 1: Reference, [c] catholic or traditional religion follower, SNNPR: South Nation Nationality peoples region, B/Gumiz: Benshangul-Gumz region

[34], Central zone Tigray (41%) [17], Slum resident in Addis Ababa (50.3%) [47], and South Gondar (47.5%) [48]. It was also lower than studies done in Southern Ghana (57%) [49], and Nepal (70%) [50]. Moreover, it was slightly lower than a study conducted in Debre-Berhan town, Ethiopia (40.6%) [40]. The variation could be due to the difference in the study settings (in which the previous studies were done in a single health institution with a small sample size) and study populations; in which most of the participants in the previous studies were urban dwellers whereas more than two-thirds (69.3%) of the mothers who participated in the current study were rural residents. So, the lower prevalence of early initiation of ANC visits reported in this study might be explained by women who lived in rural areas are less likely to have a nearby health facility which in turn exposed to other extra costs for transportation service as well as lack of availability of means of transportations. As a result, they fail to attain ANC services timely. In addition, the other possible justification might be the difference in the operational definition used to classify the outcome variable of early initiation of ANC visit. Most of the previous studies defined "early ANC visits" if the women attend ANC visits within the first 16 weeks of gestations, but, we used 12 weeks as the uppercut of point to classify as early initiation of ANC visits as recommended by WHO [51].

The study revealed that women who attended higher education were two times more likely to start their first ANC visit as compared to those women who didn't attend formal education. This finding was supported by studies conducted in Ethiopia [20, 32, 40, 52, 53], Myanmar [54], Ghana [49], Nigeria [55], Northern Uganda [56], and Sub-Saharan Africa [33]. This is explained by the fact that educated women are more economically independent, employed, have good levels of knowledge towards the benefits of attending ANC visits, the appropriate timing when it is started, and the negative consequences related to delayed initiation of ANC visits than those women who did not attend formal education [57]. Moreover, educated women might have a high possibility of having information and might have a decision-making ability on their health as well as their unborn fetus and give more attention to their health to attain the highest standards of health for themselves as well as their unborn fetus. As a result, the woman who has being educated were more likely to attend ANC visits early than woman who didn't attend formal education.

The odds of early initiations of ANC visits and household wealth status were positively associated. The odds of early initiation of ANC visits were 1.8, 1.86, and 2.3 times more likely among women who belong to the household wealth status of medium, richer, and richest as compared to women who belong to the poorest wealth status, respectively. The result was in agreement with studies conducted in Ethiopia [20, 53, 58], Ghana [49], Cameroon [59], and Sub-Saharan Africa [33]. Even though maternal health services (i.e. antenatal care, skilled birth attendants, postnatal care, and immunizations services) are delivered free of charge for all women in Ethiopia at government health facilities, services fees at private health facilities, and non-services related costs like transportation fees [60, 61] are unacceptably high. Moreover, most women are obliged to experience a long waiting time to get the services, going a long distance to and from health facilities. Such types of indirect costs are interrelated with the women's daily life of which they might go to a farm, market, office, and other workplaces to gain money to cover their daily living. As a result, women belonging to middle, richer, and richest households will be more likely to attend ANC visits early as compared to women belonging to the poorest quantile.

The current study demonstrates that the odds of the utilization of early ANC visits were 13% less likely among women whose households were headed by the male as compared to their counterparts. This finding was in agreement with other study findings at which husbands' permission had a tremendous impact on the utilization and timing of ANC services [62–64]. Another study done by Mulat G., et al. noted that the odds of using ANC visits were

more likely among womens' who had autonomy on their healthcare decision-making as compared to their counterparts [65]. Therefore, the information dissemination and education about the benefit of autonomy of women and empowerment in all dimensions of life should be much strengthened, particularly in their own healthcare decision.

Consistent with studies done in Ethiopia [66], and Rwanda [67], the odds of attending an early ANC visit was 29% less likely among women who have had $\geq$ 5 family members as compared to those women who have had less than five family members. The possible justification might be the woman who belongs to large family sizes are more likely to be prone to financial deficiency, spend more time on caring for their families, and perform all other household activities than taking care of her health.

In agreement with previous studies done in Africa [20, 32, 54, 58], this study identified that the odds of early initiation of first ANC visits were less likely among women who lived in rural areas as compared to women who lived in urban areas. The possible justification could be due to women who resided in rural areas are being exposed to inadequate availability and accessibility of health facilities, and fewer chances of getting health information as compared to women who lived in urban areas.

Our findings also depicted that women who lived in the Harari region and Dire-Dawa city administration were two and three times more likely to attend early antenatal care services as compared with women who lived in the Tigray region. Whereas, women who lived in SNNPRs were 66% less likely to attend early antenatal care services as compared to their counterparts. The variation could be due to the difference in the study populations; in which only one-fifth (19%) of the total selected participants in the Tigray region were urban dwellers whereas more than half of the nominated participants in the Harari region (54.3%) and Dire-Dawa city administration (58%) were urban residents. Also, only 9.5% of the participants selected in the SNNPR was an urban resident. Therefore, the high chances of early initiation of ANC visits reported in the Harari region and Dire-Dawa city might be explained by most of the participants being from urban areas and they are more likely to have a nearby health facility, and also have available means of transportation.

## The clinical and public health implication

This study adds for the existing body of information about modified and persistent determinants that deter women from early initiation of first ANC visits. Therefore, policy makers, and planners will workup on the identified determinates to increase the early initiation of first ANC visits and to decrease the fetomaternal complications as well as bad perinatal outcomes that occurred due to missed opportunities. The Ethiopia federal minister of health in collaboration with other non governmental organizations (NGOs) should give due emphasis to women from large family sizes, women from rural areas, women with no formal education, women from a household headed by husbands, women from poorer household wealth status, and SNNPRs regions to maximize the early initiation of first ANC visits.

## Strength and limitation

The study had many strengths, for instance, it used nationality representative data of 2019 intermediate EMDHS with a large sample size, a high response rate, and high-quality data which will reduce a bias related to sampling and measurement. Moreover, we have used an appropriate statistical approach, multilevel mixed-effect analysis, to estimate the cluster effect on early initiation of first ANC visits. Yet, we would like to assure our reader that a few limitations needed to take into account. As a cross-sectional study, the exact cause-effect relationship between early initiation of ANC visits and its predictors doesn't exist, and recall bias might be

introduced. The other limitation was the study failed to assess some important variables like distance to the health facility, pregnancy intention, media exposure and covered by health insurance, which may affect the timing of first ANC visits.

## Conclusion

In Ethiopia, despite the WHO recommends to all pregnant women attend the first ANC visits within the first 12th weeks of gestation, only less than two-fifths of the women attended antenatal care within the first trimester of pregnancy. Women who attended higher education, medium wealth status, richer wealth status, richest wealth status, living in the Harari region, and Dire-Dawa city were positively associated with the early initiation of first ANC visits. However, women who were rural residents, the household headed by a male, having ≥ 5 family members, and living in SNNPRs were negatively associated with early initiation of first ANC visits. Therefore, information dissemination should be much strengthened towards the timing of early ANC attendance, its significance for good fetomaternal outcomes, and the negative consequences related to delayed initiation of ANC visits via available media outlets by giving special attention to women who didn't attend formal education, living in an urban area and belongs to the poorest wealth quantile household. In addition to the provision of ANC services freely for all pregnant women at governmental health facilities, the governmental and non-governmental organizations should try their best to steadily increase the availability and accessibility of health facilities in rural areas to optimize the timely initiation of ANC uptake. Also, ANC services should be provided at all health facilities levels consistently, particularly in rural areas and regions with fewer attendances of early ANC visits like SNNPR. Furthermore, women's autonomy and empowerment in all dimensions of life, particularly in their own healthcare decision, should need to be given due emphasis. Lastly, the future researcher will consider a qualitative study to explore more unidentified individual, community, and facility-level factors.

## Acknowledgments

The authors acknowledge the Demographic and Health Surveys center for allowing and permitting us to access the data set.

## Author Contributions

**Conceptualization:** Gossa Fetene Abebe, Melsew Setegn Alie, Desalegn Girma, Gosa Mankelkl, Ashenafi Assefa Berchedi, Yilkal Negesse.

**Data curation:** Gossa Fetene Abebe, Melsew Setegn Alie, Desalegn Girma, Yilkal Negesse.

**Formal analysis:** Gossa Fetene Abebe, Melsew Setegn Alie, Gosa Mankelkl, Yilkal Negesse.

**Investigation:** Gossa Fetene Abebe.

**Methodology:** Gossa Fetene Abebe, Desalegn Girma, Gosa Mankelkl, Yilkal Negesse.

**Software:** Gossa Fetene Abebe, Desalegn Girma, Yilkal Negesse.

**Supervision:** Melsew Setegn Alie.

**Validation:** Gossa Fetene Abebe, Gosa Mankelkl, Ashenafi Assefa Berchedi.

**Visualization:** Gossa Fetene Abebe, Melsew Setegn Alie, Desalegn Girma, Gosa Mankelkl, Ashenafi Assefa Berchedi.

**Writing – original draft:** Gossa Fetene Abebe.

**Writing – review & editing:** Gossa Fetene Abebe, Melsew Setegn Alie, Desalegn Girma, Gosa Mankelkl, Ashenafi Assefa Berchedi, Yilkal Negesse.

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
