## [Decision Letter · Decision Letter 0]

8 Sep 2022

PONE-D-22-13822A multilevel mixed-effect analysis of determinants of early initiation of first antenatal care visit in Ethiopia; Using the recent 2019 Ethiopia mini-demographic health surveyPLOS ONE

Dear Dr. Abebe,

Thank you for submitting your manuscript to PLOS ONE. After careful consideration, we feel that it has merit but does not fully meet PLOS ONE’s publication criteria as it currently stands. Therefore, we invite you to submit a revised version of the manuscript that addresses the points raised during the review process.

Please note that we have only been able to secure a single reviewer to assess your manuscript. We are issuing a decision on your manuscript at this point to prevent further delays in the evaluation of your manuscript. Please be aware that the editor who handles your revised manuscript might find it necessary to invite additional reviewers to assess this work once the revised manuscript is submitted. However, we will aim to proceed on the basis of this single review if possible. The reviewer has identified a need to provide a clearer explanation of the rationale and motivation for your study, and to clarify aspects of your methods and statistical analyses. Please respond carefully to these and the other points raised by the reviewer when preparing your revisions.

We look forward to receiving your revised manuscript.

Kind regards,

Jamie Males

Editorial Office

PLOS ONE

Journal Requirements:

Reviewers' comments:

Reviewer's Responses to Questions

**Comments to the Author**

1. Is the manuscript technically sound, and do the data support the conclusions?

Reviewer #1: Yes

2. Has the statistical analysis been performed appropriately and rigorously? 

Reviewer #1: Yes

3. Have the authors made all data underlying the findings in their manuscript fully available?

Reviewer #1: Yes

4. Is the manuscript presented in an intelligible fashion and written in standard English?

Reviewer #1: No

5. Review Comments to the Author

Reviewer #1: General view:

I would say the authors have submitted an important piece of work in its field of relevance. However, I have few suggestions for the authors to improve this manuscript.

Title

The topic is unattractive. I suggest modification to make the title catchier, attractive and simple.

The title has some repetitions. It should be only multilevel not multilevel mixed effect

The authors said “2019 Ethiopia mini demographic health survey” which is not the right name of survey

Suggested Title “Determinants of early initiation of first antenatal care visit in Ethiopia based on the 2019 Ethiopia mini-demographic and health survey: A multilevel Analysis”

Abstract

Method: The authors should be more specific and "in order" in describing the methods. Determinants, associated factors, risk factors, please be specific throughout the document. which one do you think is appropriate considering your study design?

“A multilevel mixed logistic regression model” needs to rewritten in a correct way

Introduction

There are a lot of editorial errors in this section needs to be rewritten.

The author stated that “However, those previous studies in Ethiopia were conducted in specific areas with small sample sizes, mainly facility-based and were not nationally representative.” Is this really the case? There are many previous studies conducted with sample sizes and nationally representative (Based on EDHS 2016 may be one example). Hence, I suggest the authors to elaborate more as to why this study is important and what will be the novel contribution of this study.

The gap to this study should be clear. Thus, authors need to convince readers that this is additional information.

Determinant or factors associated please be consistent with this terminology throughout your document. Is it possible to use both terminologies interchangeably?

Methods

The author stated “2019 intermediate Ethiopian mini demography and health survey” is this the correct name of the survey? Please be consistent throughout the document

The way you define your source and study population were seeming incorrect. Please re define your source and study population.

The author stated “in the bivariable two-level binary logistic regression were candidates for multivariable multilevel logistic regression analysis.” Two-level or multi-level you wrote as the two were different and needs to be re-written

Sample size determination was not clear. On what criteria does the final sample size was considered.

Results

Add horizontal line in table, because of your data type it is difficult to see which number corresponding to which variable currently

There are a lot of editorial errors in your result. For instance, (the age range of 20-29 (55.2%) years old., 2,691 (91.7) were married, etc) please work on that

Discussion

The authors should discuss something on the existing policy or programme that target ANC services, and how these findings can be useful for improving the existing programme.

The authors should explore more on the relationship between variable. The authors should also show the implications of the results in more informative and practical way.

The authors should also follow guideline of the journal for in text and list of references

The manuscript has several grammatical and typological flaws that hamper its readability.

Align with this journal manuscript submission guideline

6. PLOS authors have the option to publish the peer review history of their article (what does this mean?). If published, this will include your full peer review and any attached files.

Reviewer #1: **Yes: **Temam Beshir Raru

---

## [Author Response · Author response to Decision Letter 0]

30 Oct 2022

Author response to reviewers

Dear Editor and Reviewers,

Thank you very much for your email dated 8th September 2022 incorporating the insight of the editor and reviewer’s comments. We, authors, would like to express our gratitude to you for the insightful and constructive review that has led to the great improvement of our paper entitled “Multilevel analysis of determinants of early initiation of first antenatal care visit in Ethiopia; using the recent 2019 Ethiopia mini-demographic health survey”. We declared that all the data underlying the results presented in the study are publicly available from the Measure DHS website: http://www.dhsprogram.com. We have carefully reviewed the comments given by the reviewer and revised the manuscript accordingly. Our responses are given in a point-by-point manner for the reviewers' comments using the Author response to reviewer form. If you have any concerns to be addressed, we are happy to consider.

Best regards!

Version 1: PONE-D-22-13822

Date: 10/9/2022

Academic editor comments and a respective author response 

Editor comment 1: Please ensure that your manuscript meets PLOS ONE's style requirements, including those for file naming. The PLOS ONE style templates can be found at https://journals.plos.org/plosone/s/file?id=wjVg/PLOSOne_formatting_sample_main_body.pdf and https://journals.plos.org/plosone/s/file?id=ba62/PLOSOne_ formatting_ sample_ title_authors_affiliations.pdf

Author’s Response: Thanks very much for this comment. The whole part of the manuscript was updated as per the PLOSE ONE style templates. 

Editor comment 2. We note that you have stated that you will provide repository information for your data at acceptance. Should your manuscript be accepted for publication, we will hold it until you provide the relevant accession numbers or DOIs necessary to access your data. If you wish to make changes to your Data Availability statement, please describe these changes in your cover letter and we will update your Data Availability statement to reflect the information you provide

Author’s Response: Thanks very much for this constructive comment. The comment has been accepted and revision has been made to the “Data availability statement” (See on the cover letter above).

Editor comment 3. We note that the grant information you provided in the ‘Funding Information’ and ‘Financial Disclosure’ sections do not match. When you resubmit, please ensure that you provide the correct grant numbers for the awards you received for your study in the ‘Funding Information section.

Author’s Response: Thanks a lot dear editor for this insightful comment. We apologize dear editor for the inconvenience that happened when editing the ‘Funding Information’ and ‘Financial Disclosure’. The suggestion of the editor has been fully accepted and corrections have been made in the funding and role of the funder as the author(s) received no specific funding for this work. So the role of the funder is “not applicable”. 

Reviewer #1 comments and an author response

Comments

General view: 

I would say the authors have submitted an important piece of work in its field of relevance. However, I have few suggestions for the authors to improve this manuscript.

Title

Comment 1: The topic is unattractive. I suggest modifications to make the title catchier, attractive and simple. The title has some repetitions. It should be only multilevel not multilevel mixed effect

The authors said “2019 Ethiopia mini demographic health survey” which is not the right name of survey. Suggested Title “Determinants of early initiation of first antenatal care visit in Ethiopia based on the 2019 Ethiopia mini-demographic and health survey: A multilevel Analysis”.

Author’s Response: Thanks very much, dear reviewer, for this insightful comment as well as the suggestion. The comment has been accepted and the title has been updated as suggested (See page 1, lines 1-3). 

Abstract 

Comment 2: Method: The authors should be more specific and "in order" in describing the methods. Determinants, associated factors, risk factors, please be specific throughout the document. which one do you think is appropriate considering your study design?

Author’s Response: Thanks very much for these constructive comments. The comments have been accepted and corrections have been made in the whole document by “Determinants”. 

 Comment 3: “A multilevel mixed logistic regression model” needs to be rewritten in the correct way.

Author’s Response: Thanks very much dear reviewer for this insightful comment. The comment has been accepted and corrected as “A multilevel mixed-effects logistic regression model” (See page 2, line 31).

Introduction

Comment 4: There are a lot of editorial errors in this section needs to be rewritten.

The author stated that “However, those previous studies in Ethiopia were conducted in specific areas with small sample sizes, mainly facility-based and were not nationally representative.” Is this really the case? There are many previous studies conducted with sample sizes and nationally representative (Based on EDHS 2016 may be one example). Hence, I suggest the authors to elaborate more as to why this study is important and what will be the novel contribution of this study. The gap to this study should be clear. Thus, authors need to convince readers that this is additional information.

Author’s Response: Thanks very much dear reviewer for these constructive comments. The comments have been accepted and corrections have been made (See page 6, lines 124-132).

Comment 5: Determinants or factors associated please be consistent with this terminology throughout your document. Is it possible to use both terminologies interchangeably?

Author’s Response: Thanks very much dear reviewer for this insightful comment. The comment has been accepted and correction has been made throughout the document by “Determinants”. 

Methods

Comment 6: The author stated, “2019 intermediate Ethiopian mini demography and health survey” is this the correct name of the survey? Please be consistent throughout the document.

Author’s Response: Thanks very much dear reviewer for this insightful comment. The comment has been accepted and corrected as “2019 Ethiopia mini-demographic and health survey” (See page 7, line 144). 

Comment 7: The way you define your source and study population were seeming incorrect. Please re define your source and study population. 

Author’s Response: Thanks very much dear reviewer for this insightful comment. The comment has been accepted and an amendment has been made (See page 7, lines 159-163). 

Comment 8: The author stated “in the bivariable two-level binary logistic regression were candidates for multivariable multilevel logistic regression analysis.” Two-level or multi-level you wrote as the two were different and needs to be re-written

Author’s response: Thanks very much dear reviewer for this comment. The comment has been accepted and correction has been made (See page 12, line 236). 

Comment 9: Sample size determination was not clear. On what criteria does the final sample size was considered.

Author’s response: Thanks very much dear reviewer for this question. In our study, the sample size was determined based on the eligibility criteria set by considering the primary intentions of the study as a pillar (i.e. To determine the magnitude of early initiation of the first ANC visit and its determinants). In the 2019 EMDHS, a total of 8,885 reproductive-age women were selected and interviewed. Of those women, only 3,979 women gave birth in the last five years preceding the survey. Of those women who gave birth, 1,044 study participants didn’t fulfill the inclusion criteria and they were excluded. Lastly, only 2,935 reproductive-age women who fulfill the inclusion criteria were enrolled in our study (See Fig 1). 

Results

Comment 10: Add horizontal line in table, because of your data type it is difficult to see which number corresponding to which variable currently.

Author’s response: Thanks very much dear reviewer for this comment. The comment has been accepted and corrected as suggested (See all tables).

Comment 11: There are a lot of editorial errors in your result. For instance, (the age range of 20-29 (55.2%) years old, 2,691 (91.7) were married, etc) please work on that

Author’s response: Thanks very much for this insightful comment. We apologize dear reviewer for the error committed during the edition stage. We accept the comment and appropriate correction has been made (See page 13, line 263)

Discussion

Comment 12: The authors should discuss something on the existing policy or programme that target ANC services, and how these findings can be useful for improving the existing programme.

Author’s response: Thanks very much for this comment and suggestion. The comment has been accepted and correction has been made as recommended (See page 19, lines 334-339). 

Comment 13: The authors should explore more on the relationship between variable. The authors should also show the implications of the results in more informative and practical way.

Author’s response: Thanks very much for this comment. The comment has been accepted and revision has been made accordingly (See pages 23-24, lines 428-436). 

Comment 14: The authors should also follow guideline of the journal for in text and list of references

Author’s response: Thanks very much dear reviewer for this insightful comment. The comment has been accepted. We used the guideline of the journal for in-text and a list of references and revision has been made accordingly. 

Comment 15: The manuscript has several grammatical and typological flaws that hamper its readability. 

Author’s response: Thanks very much for this constructive comment. We accept the comment and revisions have been made to the whole document to make it grammatically and typologically clear. 

Comment 16: Align with this journal manuscript submission guideline.

Author’s response: Thanks very much dear reviewer for this constructive comment. The comment has been accepted and alignment has been made to the journal manuscript guideline. 

I reserve any further comments until the authors have substantially revised or justified their comments.

---

## [Decision Letter · Decision Letter 1]

27 Dec 2022

PONE-D-22-13822R1A multilevel mixed-effect analysis of determinants of early initiation of first antenatal care visit in Ethiopia; Using the recent 2019 Ethiopia mini-demographic health surveyPLOS ONE

Dear Dr. Abebe,

Thank you for submitting your manuscript to PLOS ONE. After careful consideration, we feel that it has merit but does not fully meet PLOS ONE’s publication criteria as it currently stands. Therefore, we invite you to submit a revised version of the manuscript that addresses the points raised during the review process.

We look forward to receiving your revised manuscript.

Kind regards,

Demisu Zenbaba Heyi, MPH

Academic Editor

PLOS ONE

Journal Requirements:

Reviewers' comments:

Reviewer's Responses to Questions

**Comments to the Author**

1. If the authors have adequately addressed your comments raised in a previous round of review and you feel that this manuscript is now acceptable for publication, you may indicate that here to bypass the “Comments to the Author” section, enter your conflict of interest statement in the “Confidential to Editor” section, and submit your "Accept" recommendation.

Reviewer #1: (No Response)

2. Is the manuscript technically sound, and do the data support the conclusions?

Reviewer #1: Yes

3. Has the statistical analysis been performed appropriately and rigorously? 

Reviewer #1: Yes

4. Have the authors made all data underlying the findings in their manuscript fully available?

Reviewer #1: Yes

5. Is the manuscript presented in an intelligible fashion and written in standard English?

Reviewer #1: Yes

6. Review Comments to the Author

Reviewer #1: There are a lot of editorial errors in this section needs to be rewritten.

for instance, “As far as the researcher search of pieaces of literature concerned, the is no research assessing the determinants of early initiation…”

I suggest the authors to elaborate more as to why this study is important and what will be the novel contribution of this study.

7. PLOS authors have the option to publish the peer review history of their article (what does this mean?). If published, this will include your full peer review and any attached files.

Reviewer #1: **Yes: **Temam Beshir Raru

While revising your submission, please upload your figure files to the Preflight Analysis and Conversion Engine (PACE) digital diagnostic tool, https://pacev2.apexcovantage.com/. PACE helps ensure that figures meet PLOS requirements. To use PACE, you must first register as a user. Registration is free. Then, login and navigate to the UPLOAD tab, where you will find detailed instructions on how to use the tool. If you encounter any issues or have any questions when using PACE, please email PLOS at figures@plos.org. Please note that Supporting Information files do not need this step.<quillbot-extension-portal></quillbot-extension-portal>

---

## [Author Response · Author response to Decision Letter 1]

2 Jan 2023

Author response to reviewers

Dear Editor and Reviewers,

Thank you very much for your email dated 27th December 2022 incorporating the insight of the editor and reviewer’s comments. We, the authors, would like to express our gratitude to you for the insightful and constructive review that has led to the great improvement of our paper entitled “Multilevel analysis of determinants of early initiation of first antenatal care visit in Ethiopia; using the recent 2019 Ethiopia mini-demographic health survey”. We have carefully reviewed the comments given by the reviewer and revised the manuscript accordingly. Our responses are given in a point-by-point manner for the reviewers' comments using the Author’s response to the reviewer form. If you have any concerns to be addressed, we are happy to consider them.

Best regards!

Version 1: PONE-D-22-13822R1

Date: 12/29/2022

Reviewer #1 comments and an author response

Comments

Reviewers' comment #1: There are a lot of editorial errors in this section that needs to be rewritten. For instance, “As far as the researcher search of pieces of literature concerned, the is no research assessing the determinants of early initiation…” I suggest the authors to elaborate more as to why this study is important and what will be the novel contribution of this study.

Author’s Response: Thank you very much dear reviewer for this valuable comment. We, the authors, accepted the comment and the correction has been made accordingly as indicated by the reviewer (See pages 6-7, lines 121 – 138).

---

## [Editor Report · Decision Letter 2]

17 Jan 2023

Determinants of early initiation of first antenatal care visit in Ethiopia based on 2019 the Ethiopia mini-demographic and health survey: A multilevel Analysis

PONE-D-22-13822R2

Dear Dr. Abebe,

We’re pleased to inform you that your manuscript has been judged scientifically suitable for publication and will be formally accepted for publication once it meets all outstanding technical requirements.

Kind regards,

Demisu Zenbaba Heyi, MPH

Academic Editor

PLOS ONE

Additional Editor Comments (optional):

accept

Reviewers' comments:

<quillbot-extension-portal></quillbot-extension-portal>

---

## [Editor Report · Acceptance letter]

24 Feb 2023

PONE-D-22-13822R2 

Determinants of early initiation of first antenatal care visit in Ethiopia based on the 2019 Ethiopia mini-demographic and health survey: A multilevel Analysis 

Dear Dr. Abebe:

I'm pleased to inform you that your manuscript has been deemed suitable for publication in PLOS ONE. Congratulations! Your manuscript is now with our production department. 

Kind regards, 

on behalf of

Dr. Demisu Zenbaba Heyi 

Academic Editor

PLOS ONE